

# Using Aircraft Measurements to Characterize Subgrid-Scale Variability of Aerosol Properties Near the ARM Southern Great Plains Site

Jerome D. Fast[1], David M. Bell[1,2], Jiumeng Liu[1,3], Fan Mei[1], Georges Saliba[1,4], John E. Shilling[1], Kaitlyn Suski[1,5], Jason Tomlinson[1], Jian Wang[6], Rahul Zaveri[1], Alla Zelenyuk[1]

[1]Pacific Northwest National Laboratory, Richland, Washington, USA
[2]Paul Scherrer Institute, Villigen, Switzerland
[3]School of Environment, Harbin Institute of Technology, Harbin, China
[4]California Air Resources Board, Sacramento, California, USA
[5]Juul Labs, San Francisco, California, USA
[6]Washington University in Saint Louis, Saint Louis, Missouri, USA

*Correspondence to:* Jerome D Fast (jerome.fast@pnnl.gov)

**Abstract:** Complex distributions of aerosol properties evolve in space and time as a function of emissions, new particle formation, coagulation, condensational growth, chemical transformation, phase changes, turbulent mixing and transport, removal processes, and ambient meteorological conditions. The ability of chemical transport models to represent the multi-scale processes affecting the lifecycle of aerosols depends on their spatial resolution since aerosol properties are assumed to be constant within a grid cell. Subgrid-scale-dependent processes that affect aerosol populations could have a significant impact on the formation of particles, their growth to cloud condensation nuclei (CCN) sizes, aerosol-cloud interactions, dry deposition and rainout, and hence their burdens, lifetimes, and radiative forcing. To address this issue, we characterize subgrid-scale variability in terms of measured aerosol number, size, composition, and CCN concentrations made by repeated aircraft flight paths over the Atmospheric Radiation Measurement (ARM) program's Southern Great Plains (SGP) site during the Holistic Interactions of Shallow Clouds, Aerosols and Land Ecosystem (HI-SCALE) campaign. Subgrid variability is quantified in terms of both normalized frequency distributions and percentage difference percentiles using grid spacings of 3, 9, 27, and 81 km that represent those typically used by cloud-system resolving models as well as the current and next generation climate models. Even though the SGP site is a rural location, surprisingly large horizontal gradients in aerosol properties were frequently observed. For example, 90% of the 3, 9, and 27 km cell mean organic matter concentrations differed from the 81 km cell around the SGP site by as much as ~46%, large spatial variability in aerosol number concentrations and size distributions were found during new particle formation events, and consequently 90% of the 3, 9, and 27 km cell mean CCN number concentrations differed from the 81-km cell mean by as much as ~38%. The spatial variability varied seasonally for some aerosol properties, with some having larger spatial variability during the spring and others having larger variability during the late summer. While measurements at a single surface site cannot reflect the surrounding variability of aerosol properties at a given time, aircraft measurements that are averaged within an 81-km cell were found to be similar to many, but not all, aerosol properties measured at the ground SGP site. This analysis suggests that it is reasonable to directly compare most ground SGP site aerosol measurements with coarse global climate model predictions. In addition, the variability quantified by the aircraft can be used as an uncertainty range when comparing the surface point measurements to model predictions that use coarse grid spacings.



## 1 Introduction

Complex distributions of aerosols evolve in space and time as a function of emissions, turbulent mixing and transport, coagulation, chemical transformation, phase changes, ambient temperature and humidity, cloud properties, and removal processes. These processes affect the number concentration, size

50 distribution, chemical composition, mixing state, and microphysical properties of aerosols that ultimately determine the ability of aerosols to alter the radiation budget (e.g. Kaufman et al. 2002) and act as cloud or ice nucleating particles (Petters and Kreidenweis 2007; DeMott et al. 2010). As aerosols are entrained into clouds, complicated aerosol-cloud interactions perturb cloud hydrometeors, albedo, growth, dissipation, lifetime, and precipitation that influence climate (Twomey 1974; Albrecht 1989; Rosenfeld et

55 al. 2014). The ability of models to represent the multi-scale processes affecting the aerosol lifecycle depends not only on their ability to capture the important chemical and microphysical process, but also on their spatial resolution. Models often assume aerosol properties and meteorological conditions that influence aerosol evolution to be constant within a grid cell. As a result, coarse grid spacings currently used by Earth system models are not likely to resolve the large spatial variability of aerosol properties that

60 are observed in many regions of the world (e.g. Anderson et al. 2003). While decades of work have gone into developing subgrid treatments of clouds in models (e.g. Arakawa and Schubert 1974), characterizing and treating subgrid variability of aerosol processes have received far less attention. Subgrid-scale processes that affect aerosol populations could have a significant impact on the formation of particles, their growth to cloud condensation nuclei (CCN) sizes, aerosol-cloud interactions, dry deposition, wet

65 scavenging, and hence their burden, lifetimes, and radiative forcing (e.g., Wang 2007).

Modeling studies have explored subgrid-scale variability of aerosol properties and their effects on atmospheric forcing via their interactions with radiation and clouds by comparing the model predictions using different grid spacings. For example, climate model simulations in Ekman and Rodhe (2003) quantified the impact of anthropogenic sulfate on cloud albedo using grid spacings of 2.0 and 0.4 degrees.

70 They found that the global mean indirect radiative forcing differed by only 7% and there was no difference in the temperature response; however, they did note larger regional differences in aerosol radiative forcing and atmospheric response. Qian et al. (2011) and Gustafson et al. (2011) used the chemistry version of the Weather Research and Forecasting model (WRF-Chem) with 75, 15, and 3 km grid spacings to determine differences in representing aerosol variability over Mexico and found that

75 neglecting small-scale variations in aerosols lead to biases in shortwave radiative forcing as large as 30%. Wainwright et al. (2012) performed simulations using the chemistry version of the Goddard Earth Observing System model (GEOS-Chem) with grid spacings of 4, 2, and 0.5 degrees and found that predicted secondary organic aerosol (SOA) concentrations depended significantly on model resolution because of non-linear effects associated with SOA partitioning, lifetime, and precursor emissions. By

80 using both a global climate model and WRF-Chem, Ma et al. (2014) showed that grid spacings much smaller than typical global climate models were needed to reproduce the observed black carbon plumes transported over the Pacific Ocean. Weigum et al. (2016) performed identical simulations using WRF-Chem except with grid spacings of 80 and 10 km and found that the coarser simulation of aerosol optical depth (AOD) was underpredicted by 13% and CCN concentration was overpredicted by 27%.

85 These studies show that models are useful tools to investigate variability in aerosol properties and aerosol-radiation-cloud interactions associated with a range of grid spacings; nevertheless, measurements are critical to quantify those variabilities observed in the atmosphere. While there are surface monitoring networks in North America, Europe, and parts of Asia that collect PM2.5 and PM10 measurements, the density of stations is often insufficient to fully characterize spatial variabilities in aerosol mass. In urban

90 areas, emissions of aerosols and their precursors vary significantly so that many monitors are needed to



characterize variability in aerosol mass and a single monitor is not likely to be representative of a large area (e.g. Schutgens et al. 2017). The spatial representativeness of aerosol measurements is likely larger over rural areas, but the density of monitors also decreases. The distance between monitors varies between tens of kilometers to a few hundred kilometers apart so that regional variability in aerosol mass may be underestimated in rural regions as well. Asher et al. (2022) recently described the spatial variability in aerosol number concentrations across a rural region around the Atmospheric Radiation Measurement (ARM) program's Southern Great Plains (SGP) site in northern Oklahoma using measurements from a network of seven Portable Optical Particle Spectrometers (POPS). They found that over a five-month period concentration of aerosols between 140 and 2500 nm in diameter varied by 10 to 20% among the sites.

Satellite measurements have more complete spatial coverage than operational monitoring networks during clear-sky conditions, but they provide column integrated quantities (e.g. AOD) and do not quantify more complex aerosol properties such as number, composition, and size distribution as a function of altitude. Therefore, aircraft platforms that deploy instrumentation capable of characterizing a wide range of aerosol properties have been used to fill in this data gap. Many research aircraft missions have been conducted over the past several decades to quantify variations in aerosol properties in many regions throughout the world. For example, Weigum et al. (2012) used the global HIAPER Pole-to-Pole Observations (HIPPO) over the Pacific Ocean and suggested that the horizontal variations in black carbon plumes could not be resolved by climate models at that time. Many aircraft field campaigns are designed to be surveys over global (e.g., Wofsy et al. 2011; Brock et al. 2019) to regional (e.g., Toon et al. 2016) scales. For these surveys, a specific region is sampled once that provides a "snapshot" of the variability that may or may not be representative of that region over a long period. Fewer aircraft campaigns have been conducted with repeated flight transects over a region. Those campaigns are often conducted over urban, areas such as Houston (e.g., Parrish et al. 2009) and Los Angeles (Ryerson et al. 2013), to better understand aerosol formation and removal mechanisms, rather than being used to characterize spatial variability of aerosol properties.

In contrast to Asher et al. (2022) that quantified the spatial variability of aerosol number concentrations and size distributions using surface measurements, we address this issue characterizing the subgrid-scale variability in terms of measured aerosol number, size, composition, and CCN concentrations using aircraft measurements from the HI-SCALE campaign (Fast et al. 2019) near the ARM SGP site. Repeated flights were conducted over a predominantly rural area ~160 km wide in north-central Oklahoma during the spring and late summer of 2016. Subgrid variability of aerosol properties is quantified for multiple grid spacings, ranging from those typically used by current cloud-system resolving models as well as current and next generation climate models. As will be shown later, surprisingly large horizontal gradients in aerosol properties were frequently observed even though the SGP site is a rural location. As expected, the smaller grid spacings capture more of the overall variability in aerosol properties; however, there is still substantial subgrid-scale variability using grid spacings of 3 km. We also find a seasonal dependence on subgrid-scale variability for some aerosol properties, with more variability during the spring for some properties and more variability during late summer for other properties. These seasonal differences are due, in part, to changes in variable precursor emissions of secondary organic aerosols (SOA) and SOA formation rates. We also compare the aircraft measurements in the boundary layer with those collected at a surface site to determine the representativeness of the surface measurements.



## 2 HI-SCALE Aircraft Measurements

Measurements of aerosol number, size, and composition were collected by the DOE's Gulfstream-1 (G-1, Schmid et al. 2014) aircraft over north-central Oklahoma near the ARM SGP facility (Sisterson et al. 2016) during the Holistic Interactions of Shallow Clouds, Aerosols and Land Ecosystems (HI-SCALE) campaign (Fast et al. 2019). Two Intensive Observational Periods (IOPs) were conducted by the G-1 aircraft, one in the spring between April 24 and May 21 and the other in the late summer between August 28 and September 24. 17 and 21 flights were conducted during IOPs 1 and 2, respectively. One flight was conducted per day, except on five days during IOP 2 that had two flights per day. All flight paths during IOP 2 are shown in Fig. 1, which were similar to those during IOP 1. The G-1 was based out of Bartlesville, Oklahoma, east of the SGP site. 66% and 59% of the flight time during spring and summer, respectively, occurred within 100 km of the SGP site's Central Facility. Since the observed winds during the IOP periods are frequently from the south to southeast as shown by wind roses in Fig. S1, measurements were collected along two transects upwind of the SGP site on several days.

Table 1 lists the instrumentation and measurements of aerosol number, size, and composition used in this study to quantify spatial variability of those quantities. Additional instrumentation deployed on the G-1 and the resulting measurements are described in Fast et al. (2019). Two TSI Condensation Particle Counters (CPC models 3025 and 3010) were deployed to measure number concentrations for particle diameters greater than 3 and 10 nm. The Fast Integrated Mobility Spectrometer (FIMS, Wang et al. 2018) measured number distribution for particle diameters between 9 and 426 nm. CCN number concentrations at two supersaturations (0.24 and 0.46%) were measured by a Droplet Measurement Technology (DMT) CCN counter. An Aerodyne Time-of-Flight Aerosol Mass Spectrometer (AMS, DeCarlo et al. 2006) provided bulk aerosol composition concentrations and mass concentrations for nonrefractory organic matter (OM), sulfate ($SO_4$), nitrate ($NO_3$), and ammonium ($NH_4$) for particle diameters < 1 μm vacuum aerodynamic diameter particles. The average concentrations of OM, $SO_4$, $NO_3$, and $NH_4$ for each flight is given in Tables S1 and S2. Detailed information on aerosol composition is provided by miniSPLAT (Zelenyuk et al. 2015), a single particle mass spectrometer that characterizes composition of individual particles with vacuum aerodynamic diameters between 50 nm and 2 μm. The observed aerosol mixing state during HI-SCALE is often complex (Fast et al. 2019), which is different than assumptions of internal or external mixtures of aerosol composition commonly used by climate models. To reduce the complexity of aerosol mixtures, particles sampled by miniSPLAT are divided into 11 classes as described in Table 2. Most of the classes have mixtures of different types of organics. miniSPLAT is also able to characterize refractory particles containing soot and dust that the AMS cannot detect. Two classes (Org_amins, Pyr) are not analyzed in this study since relatively few of those particles were observed during HI-SCALE. Particle classes were determined at 6 min intervals which has implications on spatial variability that will be described later. The average particle class fractions for each flight is given In Tables S3 and S4.

Given that the G-1 flight speed was 100 m s⁻¹, each measurement at 1-s intervals from the CPC, FIMS, and CCN instruments represents average conditions over 100 m. In contrast, each measurement from the AMS and miniSPLAT average conditions over 1.3 km and 6 km, respectively.

At the Central Facility, similar instrumentation as on the G-1 aircraft was deployed to obtain continuous measurements of aerosol number, size, composition, and CCN concentrations. We also use Doppler lidar measurements from the Central Facility, E32, E37, E39, and E41 sites denoted by the white circles in Fig. 1. The Doppler lidar provides vertical profiles of wind speeds and directions, but vertical gradients in the signal-to-noise (SNR) ratio are also used to determine the convective boundary layer height as a function



of time during the day. In this study, we use these heights to determine which aircraft flight legs are conducted within and above the boundary layer.

Emission rates of primary organic aerosol (POA) at the surface and sulfur dioxide ($SO_2$) above the surface from the 2011 EPA National Emissions Inventory (NEI) are also shown in Fig. 1a to illustrate the spatial distribution of anthropogenic sources of aerosols and aerosol precursors in the region in relation to the G-1 flight paths. The Oklahoma City, Tulsa, and Wichita metropolitan areas have the largest urban emission rates in the region. There are also several large sources of $SO_2$ located in rural areas, such as the Redrock Power Plant and the Ponca City Refinery near the SGP site. The G-1 aircraft frequently observed

peak concentrations of $SO_2$ and aerosol number close to these sources. As described in Liu et al. (2021), the AMS measurements indicated that organic matter made up the largest fraction of aerosol mass during both IOPs and biogenic trace gas emissions likely contributed to a significant fraction of the observed organic aerosol mass. As shown in Fig. 1b, the largest sources of isoprene emissions are located to the south and east of the SGP site, and southeasterly winds would transport fresh biogenic SOA towards the

SGP site. These isoprene rates are based on the MEGAN model (Guenther et al. 2012) and the spatial distribution coincides with the density of broadleaf trees (primarily Oak) that emit isoprene and the emission rates depend on meteorological conditions which are different between IOPs 1 and 2.

As described later, miniSPLAT measurements frequently detected particles with biomass burning markers during both IOPs. On a couple of days during IOP 1 (May 6 and 7, Table S3), biomass burning aerosols

comprised a large fraction of the total aerosol number. Biomass burning emissions from the Fire INventory from NCAR (FINN, Wiedinmyer et al. 2011) in Fig. S2 indicate that the number of fires and emission rates were relatively low around the SGP site and the number of fires were even lower during IOP 2. However, coupling air mass trajectories shown in Liu et al. (2021) with FINN suggests that fires as far north as North Dakota and southern Canada may have contributed to biomass burning aerosol during

IOP 1. During IOP 2, biomass burning aerosols were likely transported by southeasterly winds from larger fires in Arkansas, Louisiana, and southeastern Texas. Taken together, Figs. 1, S1, and S2 suggests that local anthropogenic, biogenic, and biomass burning sources likely contributed to aerosols sampled by the G-1 in addition to long-range transport.

## 3 Defining Subgrid-Scale Variability

Our objective is to compute the spatial variability of observed aerosol properties in relation to typical model grid cells that assume constant values within those grid cells. First, grid cells over the G-1 flight domain are defined with widths of 81, 27, 9, and 3 km as shown in Fig. 2. The 81 and 27 km cells represent typical grid spacings used in current climate models, while the 9 and 3 km cells represent typical grid spacings used by current weather forecast and chemical transport models. Next generation

climate models are expected to have similar spatial resolutions as current weather forecast models. Four 81-km cells encompass most of the HI-SCALE G-1 flights paths. This spatial configuration results in 36, 324, and 2196 cell for the 27, 9, and 3 km grid spacings, respectively. Mean values of aerosol number, aerosol composition, and CCN concentration are then computed for each cell, flight leg, and flight. Only data on constant flight legs are used so that effects of vertical gradients on the subgrid-scale variability

statistics are minimized. Data for flight legs within clouds are also excluded from the statistics.

For example, Fig. 3a shows the constant altitude flight legs using alternating red and blue colors and Fig. 3b shows the temporal variability of OM, $SO_4$, $NO_3$, and $NH_4$ concentrations sampled by the AMS on May 7. Horizontal gradients in OM over individual flight legs are evident by comparing Figs. 3a and 3b. In addition, the spatial variability of $SO_4$, $NO_3$, and $NH_4$ over a particular flight leg can be quite different

because of different aerosol precursor sources and secondary formation processes. The variability of the



grid cell means for the 81-km cell centered over the SGP site (Fig. 3c) is shown in Fig. 3d. Mean OM concentrations for flight legs (green dots) within the 81-km cell is between 3.21 and 3.47 µg m$^{-3}$. The range of the mean values increase for the 27 and 9 km cells; however, the range is similar for the 9 and 3 km cells. This suggests that a 9 km grid spacing captures much of the variability over the G-1 flight path.

While the 3 km cells do not provide a larger range in cell means, they do result in a larger standard deviation of OM that reflect some small-scale variability that even the 3 km cell mean does not represent. Since the AMS sampling interval of ~13 s results in only 2 to 3 samples within a single 3 km cell, we note that the AMS is already averaging out some of the variability that can be represented by instruments with 1 s sampling intervals.

It is possible that the different flight leg altitudes introduce vertical gradient variability into the horizontal gradient calculations. While it is reasonable to assume that vertical gradients of aerosols within the daytime convective boundary layer are likely to be small, there were large vertical gradients between the top of the boundary layer and the lower free troposphere. Therefore, flight legs are separated into those within and above the convective boundary layer using a combination of vertical potential temperature 235 gradients from aircraft and radiosonde measurements and boundary layer heights derived from the Doppler lidars. Flight legs at the top of the boundary layer are excluded with those within the boundary layer. Since data from the Doppler lidars is not available prior to May 3, boundary layer depth is determined solely from radiosonde and aircraft measurements for the first five flights during IOP 1.

The average boundary layer height determined from the five Doppler lidars is included in Fig. 3a to 240 separate the flight legs into two groups. While there is some variability in the boundary layer height across the SGP site, the average boundary layer height among the five lidars is similar to the height at the Central Facility on May 7. On this day, two constant altitude flight legs are at or very close to the top of the convective boundary layer and are thus excluded from the analysis of spatial variability within the boundary layer. Relatively few flight legs were conducted in the lower free troposphere or near the 245 boundary layer top; therefore, we quantify subgrid-scale variability of aerosol properties only within the boundary layer.

We next compute a normalized frequency of the variable means over all flight legs by binning OM into concentration ranges of 0.5 µg m$^{-3}$ and binning SO$_4$, NO$_3$, and NH$_4$ into concentration ranges of 0.05 µg m$^{-3}$ as shown in Fig. 4. Fig. 4a shows the same information in Fig. 3d, but in a slightly different manner. 250 As in Fig. 3d, the mean OM concentrations over the 81 km cell falls within the 3.0 – 3.5 µg m$^{-3}$ bin and the mean cell concentrations for the 27, 9, and 3 km cells have a larger range between 2.5 and 4.5 µg m$^{-3}$. Similarly, the smaller grid spacings resolve more of the spatial variability in SO$_4$, NO$_3$, and NH$_4$ (Figs. 3b – 3d).

Similar statistics on the normalized frequency can be computed for all flights and for aerosol number, 255 composition, and CCN concentration. For example, Fig. S3 depicts the normalized frequency of aerosol composition for all flights during IOP 1. However, this method will include day-to-day variations in the mean concentrations (Tables S1 and S2) which complicates quantifying overall horizontal variability during the campaign. Instead, we compute percent differences from the 81 km cell mean given by Equation (1) for each flight leg and variable within an IOP. Then, percentiles are used to express the 260 departures from the 81 km cell mean. This removes the day-to-day variability as well as temporal variability within the same flight, so that all the flight statistics can be lumped together which will be shown in the next section.

$$\% \; difference(cell) = \left[ \frac{<cell> - <81 \; km \; cell>}{<81 \; km \; cell>} \right] * 100 \qquad (1)$$



We note that there are two additional assumptions that will affect the computed spatial variability
statistics. First, the aircraft flight path does not sample a large portion of a grid cell. At a minimum, it
represents a transect across the grid cell; therefore, we assume that the spatial variability along that line is
consistent with the variability within the entire cell. This is analogous to cloud chord sampling (e.g
Barron et al. 2020; Griewank et al. 2020). However, there were some flights that had a grid pattern near
the Central Facility (Fig. 1) which would permit multiple transects across the 81 and 27 km grid cells.
Second, temporal variability is not totally removed from these calculations. Individual flights legs were
usually10 to 20 minutes in duration. Aerosol number, size distribution, and composition which influences
CCN concentration can be affected by secondary chemical formation processes over that period; however,
we assume those effects are smaller than the spatial variability present over a flight leg.

**4 Spatial Variability as a Function Grid Spacing and Season**

**4.1 Bulk Aerosol Composition**

Figure 5 shows the variability in aerosol composition for all flight legs during IOPs 1 and 2 relative to the
coarse 81 km cell around the SGP site. For OM (Fig. 5a), 50% of the 27, 9, and 3 km cell means are
within 7 - 13% of the coarse cell mean, while 90% of the 27, 9, and 3 km cells are within 28 – 46% of the
coarse cell mean. The spatial variability of $SO_4$ (Fig. 5b) is somewhat smaller than OM since 90% of the
27, 9, and 3 km cells are within 23 - 35% of the coarse cell mean. In contrast, 50% of the 27, 9, and 3 km
grid cell means for $NO_3$ (Fig. 5c) during IOP 1 are within 16 - 33% of the coarse cell mean and 90% of
the grid cell means are within 52 – 64% of the coarse cell mean, which is much larger than the spatial
variability of OM and $SO_4$. The spatial variability in $NH_4$ (Fig. 5d) is somewhat greater than $SO_4$, but less
than $NO_3$. The coarser 27 km cell means represent a large fraction of the spatial variability produced by
the 9 and 3 km cell means when all the aircraft measurements are lumped together by IOP. For all four
composition species, the 3 km cells have a somewhat broader distribution as evident by larger range
between the minimum and maximum values and relatively fewer values closer to zero than the 27 km
cells. In general, all the variability among all the 3 km cell means is within ~100% of the coarse grid cell
mean.

It is important to note that during IOP 2 the average OM concentrations were 1.3 μg m$^{-3}$ (52%) higher,
$SO_4$ concentrations were 0.5 μg m$^{-3}$ (63%) higher, and $NO_3$ concentrations were 0.16 μg m$^{-3}$ (64%) lower
than during IOP 1 as described by Liu et al. (2021) and reflected by the distribution of concentrations
shown in Fig. S4. In addition to meteorological conditions being more conducive to SOA formation
during IOP 2, local peak concentrations around the SGP site produce additional variability in OM as
reflected by the broader distribution of the absolute difference between the 3 km cell means within the
coarse cell mean as shown in Fig. S5. Similarly, the absolute difference in $NO_3$ has more local variability
around the SGP site during IOP 1. Inspection of individual IOP 1flights show that there were short
periods of relatively large $NO_3$ concentrations, suggesting there were local $NO_3$ hotspots as the G-1
passed over the SGP site. These events occurred were less frequent during IOP 2, most likely because
temperatures were significantly higher during the late summer period which would affect the temperature
dependence of $NO_3$ to gas-to-particle partitioning processes. In contrast, the distribution of the absolute
differences for $SO_4$ and $NH_4$ are similar for IOP 1 and 2.

Since Equation (1) normalizes the results by the 81 km cell mean much of the seasonality of absolute
spatial differences in OM is removed as indicated by Fig. 5a. However, Fig. 5c shows that Equation (1)
still leads to smaller spatial variability of $NO_3$ during IOP 2. For example, the 25$^{th}$ to 75th percentile range





during IOP 2 is within 8 - 19 % of the coarse cell mean, which is smaller than the 16 - 33% range during IOP 1.

Since total aerosol mass is dominated by organics, spatial variability in total mass concentrations over the SGP site will look similar to Fig. 5a in terms of percent differences of OM from the 81 km cell mean. In

terms of absolute differences, the spatial variability of total mass concentrations over the SGP site are expected to be larger during the summer IOP, as with OM shown in Fig. S5. The variations among the 27, 9, and 3 km cells as well as the seasonality for all four 81 km cells shown in Figure 2a are very similar to those in Fig. 5 (not shown).

### 4.2 Single-Particle Composition

The aerosol class information from miniSPLAT is expressed as a number fraction of the total number of characterized particles. To account for aerosol loadings, we compute sub-grid variability for the aerosol classes (Table 2) in terms of aerosol volume by multiplying the particle fraction and the total volume measured by FIMS. This assumes the same FIMS-measured size distributions for all aerosol classes, while in reality, some particles classes are likely to be more frequent at small (i.e. Soot) and large (i.e.

Dust) sizes. In addition, miniSPLAT provides critical information on aerosol composition during IOP 2 for the flights between August 30 and September 7 when the AMS was not functioning.

Fig. 6 shows the spatial variability of nine particle classes from miniSPLAT for all aircraft flights during IOPs 1 and 2 in terms of percent difference from the 81 km cell mean. The percentiles for the Sulfate_org, Nitrate_org, IEPOX_SOA (for IOP 2), Org 1, Org 2 (for IOP 2), BB_SOA, and BB classes in Figs. 6 are

similar to the statistics on bulk OM (Fig. 5) in terms of having a wider range between the 5[th] and 95[th] percentiles among the 3 km cells compared to the 27 km cells. However, the departures from the coarse cell mean for these five particle classes is larger than for bulk OM measured by the AMS. For example, 90% of the 27, 9, and 3 km cells for OM are within 28 – 46% of the coarse cell mean while 90% of these five particle classes are within 42 - 87% of the coarse cell mean. In addition, the Org1 (fresh organic rich)

and Org2 (aged organic rich) classes (Figs. 6d,e) have smaller departures from the coarse cell mean during IOP 2. 90% of the 27, 9, and 3 km cells during IOP 1 are within 61 – 81% of the coarse cell mean which shrinks to 28 – 41% during IOP 2. These differences are likely due to the relative contributions of organic-rich particles during the two IOPs, with significantly higher contributions from fresh and aged organic rich particles during IOP 2 compared to IOP 1. Many days during IOP 1 were dominated by the

sulfate-organic mixtures (Sulfate_Org) or biomass burning particles (BBOA_SOA and BB), with relatively small contributions of organic-rich particles and larger spatial variability over the SGP site. In contrast, the largest fraction of organic aerosols on most days during IOP 2 were comprised of organic-rich particle classes (Org1, Org2) that were more evenly distributed over the SGP site. The smaller fraction of the Sulfate_org particle class during IOP 2 is consistent with the lower $SO_4$ concentrations

observed by the AMS during IOP 2 compared to IOP 1.

Another aspect of the distributions of the organic classes is that some classes during IOP 1, such as Nitrate_Org and Org1, are more skewed than others. This has implications if models wish to represent subgrid-scale variability of aerosol properties in some way. This is analogous to how some cloud parameterizations (e.g. CLUBB, Golaz et al. 2002) use assumed Gaussian or double Gaussian probability

density distributions to represent subgrid-scale variability of cloud properties.

While soot is usually a small fraction of total aerosol mass, it still has an important role in direct aerosol shortwave radiative forcing. Fig. 6f shows that the 3 km cells have a larger range than the 27 km cells and 90% of the 3 km cell means within 65 – 72% of the coarse cell mean, reflecting contributions of both



local and distant anthropogenic sources to spatial variability in this predominately rural location. The
observed distribution of soot is also skewed and neglecting this skewness within coarse climate model
grid cell averages may lead to uncertainties in the predictions of aerosol absorption. The spatial variability
of fine-mode dust and its skewed distribution (Fig. 6i) is similar to that of soot and coarse climate models
grid cell averages may also lead to uncertainties in the aerosol longwave radiative forcing.

Since the particle class information from miniSPLAT was computed over 6 min intervals that represents
an average over 6 km, the 9 and 3 km cell percentiles in Fig. 6 are nearly the same. Therefore, the actual
spatial variability in aerosol mixing state may be larger than suggested by the 9 and 3 km cells in Fig. 6.

For reference, Fig. S6 shows the absolute differences between the 3 km cell means and coarse cell mean
for the nine particle class volumes during IOPs 1 and 2. The single particle measurements reveal that
aerosol composition is far more complex than the bulk AMS composition measurements, and the spatial
variability differs among the particle classes. These differences in the bulk and single particle
representations have implications on the overall hygroscopicity of particle populations and consequently
concentration of CCN and aerosol-cloud interactions.

### 4.3 Aerosol Number Concentration and Size Distribution

Figure 7 illustrates the temporal variability on the May 7 aircraft flight for the particle number
concentration from the CPC3025 and FIMS instruments along with the CCN concentration at 0.24%
supersaturation. Large spatial gradients in aerosol number concentrations were observed along the
individual flight legs with peak concentrations frequently exceeding 6000 cm$^{-3}$ (Fig. 7a). While CPC3025
measurements show that high aerosol number concentrations were observed at many locations around the
Central Facility (Fig. 7b), continuous ground measurements of aerosol size distribution at the Central
Facility (not shown) did not indicate a strong new particle formation event on this day. The
spatiotemporal variations in number concentration from FIMS were similar to those from the CPC3025,
although the concentrations were lower. Comparison of CPC3025 and FIMS concentrations show that a
large fraction of particles were smaller than 9 nm in diameter. Since the smallest particle diameter from
the CPC3010 and FIMS instruments are nearly identical, the CPC3010 concentrations are close to those
from FIMS and therefore not shown for clarity. In addition, the FIMS number concentrations for
diameters between 49 and 426 nm are shown to represent the variations in the lower portion of the
accumulation mode size range.

The normalized frequency of the mean number concentrations over 500 cm$^{-3}$ bin intervals from CPC3025
and FIMS number concentrations are shown in Figs. 7c and 7d, respectively, for the 81, 27, 9, and 3 km
cells. For the 81-km cell around the SGP site, mean CPC3025 concentrations (Fig. 7c) along the flight
legs occurred within four bins between 4500 – 6500 cm$^{-3}$. Progressively wider distributions are produced
for the smaller grid cells, so that the peak mean concentrations increase to 6500-7000 cm$^{-3}$, 7000-7500
cm$^{-3}$, and 8500-9000 cm$^{-3}$ for the 27, 98, and 3 km cell sizes, respectively. The results from FIMS (Fig.
7d) are similar to those from the CPC3025, except that the highest frequencies occur at lower
concentrations. Figure 7c shows that the 3 km cells have the broadest distribution and thus captures more
of the variability of small particles from the CPC3025, while both the 3 and 9 km cells captured much of
the variability of the larger particles measured by FIMS as shown in Fig. 7d. The range of the aerosol
number and volume distribution within the 81-km cell around the SGP site is shown in Figs. 7e and 7f,
respectively to illustrate the differences among the 18, 27, 9, and 3 km cells as a function of aerosol
diameter. The range of aerosol number (Fig. 7e) is largest for with diameters less than 50 nm on this day,


consistent with Fig. 7c. The spatial variations in the accumulation mode aerosol volume (Fig. 7f) are similar to the variability of organic aerosol mass (Fig. 4a).

These results imply that coarse climate models are not likely to represent the variability of number concentrations over land, especially on days when new particle formation processes that produce high concentrations of ultrafine particles are important.

The spatial variability of number concentration for all aircraft flights during IOPs 1 and 2 in terms of percent difference from the 81 km cell mean is shown in Fig. 8. For IOP 1, the range of the 5th and 95th percentiles for the 27, 9, and 3 km cells are very similar for the CPC3025 (Fig. 8a) and CPC3010 (Fig. 8b) measurements for diameters greater than 3 and 10 nm, respectively. 90% of the grid cells means are within 36 – 55 % and 41 – 55% of the coarse grid cell for particle diameters greater then 3 and 10 nm, respectively. Nevertheless, the range of the remaining 10% becomes progressively larger for the 27, 9 and 3 km cells. In contrast, the spatial variability is much larger and more skewed during IOP 2 with 90% of the grid cells means with 45 – 68 % and 47-64% of the coarse grid cell for diameters greater than 3 and 10 nm, respectively. Fast et al. (2019) note that 20 and 6 NPF events were observed during IOPs 1 and 2, respectively, but these results show that the overall spatial variability of number concentrations also differed between the IOPs. During IOP 1, high number concentrations were frequently observed over a large portion of the SGP site similar to the distribution on May 7 (Fig. 7a,b). Except for a couple of flights, peak number concentrations were more often isolated and usually associated with emissions from the Redrock Power Plant and the Ponca City Refinery (Fig. 1a) during IOP 2. Therefore, the smaller spatial variability represented by the 27, 9, and 3 km cells during IOP 1 (Figs. 8a and 8b) reflects the more uniformly distributed number concentrations over the 81 km cell even though the number concentrations tend to be higher during IOP 1 than during IOP 2 (Fig. S7).

Since the lowest size bin from FIMS is close to the smallest particle measured by the CPC3010, Figs. 8b and 8c are similar for the 27, 9, and 3 km cell as well as the differences between the IOPs. When the smallest particles less than 49 nm in diameter are removed, Fig. 8d shows that the spatial variability in number concentrations decrease significantly. For this size range, 90% of the 27, 9, and 3 km cells are within 23 – 34% and 27-35% of the coarse cell mean for IOPs 1 and 2, respectively. This range is comparable to the range of OM mass concentrations where most of the mass is in the accumulation mode. For reference, the absolute difference in number concentrations during IOPs 1 and 2 between the 81 and 3 km cells are shown in Fig. S8, which shows that these differences are frequently as large 4000 cm$^{-3}$ when including ultrafine aerosol sizes and 500 cm$^{-3}$ when considering accumulation mode sizes.

It is important to note that the variability expressed in Fig. 8 is larger than the 10 to 20% spatial variability in aerosol number concentrations over the SGP site reported by Asher et al. (2022). The differences between this study and Asher et al. (2022) are likely due to three factors. First, the POPS instrument measures particles as small at 140 nm which neglects Aiken and ultrafine particles that measured by the CPC and FIMS instruments. Second, the surface measurements in Asher at al. (2022) were collected during the fall and winter months between mid-October 2019 to mid-March 2020 and therefore the spatial variability may differ from aerosol populations observed during the spring and late-summer as sampled during HI-SCALE. Finally, the different methodology of quantifying spatial variability from fixed surface sites and aircraft paths is likely a contributing factor.

The previous statistics on aerosol number hide important spatial variability in aerosol number and volume distributions that have implications for CCN since it depends strongly on aerosol size (e.g., Dusek et al. 2006; Pöhlker et al. 2016; Patel and Jiang 2021). The range of aerosol number and volume distribution in Figs. 7e and 7f suggest that the Aiken and accumulation modes each consist of a single mode diameter. Aerosol number and volume distribution averaged over an 81 km cell will likely have a simple, smooth distribution. New representations of aerosol formation and growth processes in climate models that better agree with such averaged observations may be obtaining the right answer for the wrong reasons because the spatial variability of aerosol and growth processes are likely to be far more complicated. For example,



81 km mean aerosol number and volume size distributions for September 17 and May 7 are shown in Fig.
9, along with select distributions from seven 3 km cells within the 81 km cell. On September 17 during a
new particle formation event (Fig. 9a), the aerosol number concentration of the Aitken mode is about
twice that of the accumulation mode. The average Aitken mode diameter for the 81 km cell occurs at 30
nm while the Aitken mode diameters from select 3 km cells range between 20 and 40 nm, indicating
variability in the growth rate of the Aitken mode over the 81 km cell. Interestingly, the aerosol volume
size distributions from 3 km cells show a distinct bimodal distribution within the accumulation mode.
While the bimodal distribution is still visible in the 81 km cell averaged accumulation volume mode, it
appears to be less pronounced than the 3 km cell averages. Zaveri et al. (2022) found similar bimodality
progressively develop within the accumulation volume mode in aged urban air in the Amazon. They used
a detailed aerosol box model to interpret this behavior and attributed it to preferential growth of the
smaller Aitken mode particles from SOA formation at the expense of slow, bulk diffusion-limited growth
of the preexisting semisolid accumulation mode. This aerosol growth behavior, however, could not be
reproduced by the equilibrium SOA partitioning approach typically used in many chemistry-aerosol-
climate models.

On May 7 in Fig. 9b, the Aitken mode number concentrations are about half that of the accumulation
mode for the 81 km average. While the 3 km cells show that the number concentrations for particle sizes
less than 40 nm can be up to a factor of two larger than the 81 km average, these variations are much
smaller than those on September 17. In addition, the overall shape of the aerosol number distributions
from the 3 km cells tends to be similar to the 81 km average in contrast to September 17. Nevertheless,
the aerosol volume size distributions from the 3 km cells still exhibit the characteristic bimodality within
the accumulation mode, suggesting a history of rapid Aitken mode growth and bulk diffusion-limited
growth of the preexisting semisolid accumulation mode from SOA formation.

Overall, the results in Fig. 9 show important differences in aerosol number and volume size distributions
over the 81 km cell region that reflect subgrid scale variability in the new particle formation events and
the subsequent particle growth rates due to variable precursor emissions and meteorological conditions.
Thus, an equilibrium aerosol treatment used in a coarse climate model that may approximately reproduce
the average aerosol number and volume size distributions, such as the 81 km cells shown in Fig. 9, may
not work at progressively higher spatial resolutions where more variability and important features are
seen in aerosol size distributions due to variability in NPF rates and complex aerosol growth behaviors.

### 4.4 Cloud Condensation Nuclei

CCN is an important metric for climate models, since it drives aerosol-cloud interactions that influence
the Earth's radiation budget and precipitation patterns. The representations of aerosol-cloud interactions
in climate models remains highly uncertain (Boucher et al. 2013; Naik et al. 2021) because of the non-
coincident spatiotemporal variations in aerosol and cloud populations and their representation by coarse
grid spacings as well as parameterized treatments of aerosol and cloud properties.

The spatial variability of CCN from the G-1 flight legs during IOPs 1 and 2 at two supersaturations are
shown in Figs. 10. For CCN at 0.24% supersaturation during IOP 1, 50% of the 27, 9, and 3 km cell
means are within 4 – 8% of the coarse cell mean and 90% of the cell means are withing 22 – 38% of the
coarse cell mean. The remaining 10% of the 3 km cells differed by as much 100% of the coarse cell mean.
During IOP 2, the spatial variability is somewhat larger for the $25^{th}$ to $75^{th}$ percentiles which are within 6
– 11% of the coarse cell mean. However, the maximum differences with the coarse cell mean are similar
for the 27, 9, and 3 km cells. For CCN at 0.46% supersaturation, 50% of the 27, 9, and 3 km cell means
are within 5 – 15% of the coarse cell mean, which is somewhat larger than for CCN at 0.24%
supersaturation. While the percent differences among the 27, 9, and 3 km cells and between the IOPs
differ somewhat, the spatial variability for CCN at both supersaturations is similar with little skewness
and medians close to zero. Interestingly, the 5th to 95th variability in CCN concentrations among the 27,
9, and 3 km is significantly lower than the observed variability in aerosol composition and mixing state





(Fig. 6) and number concentrations (Fig. 8), especially at the 0.24% supersaturation, even though both aerosol properties determine their CCN activity. This result indicates that improved representations of aerosol formation and growth processes may results in modest improvements in CCN concentrations and variability at ambient supersaturation levels, and hence modest changes in modeled indirect radiative effects.

CCN concentrations at 0.46% supersaturation are usually higher than those at 0.24% and CCN at both supersaturations is usually higher during IOP 2(Tables S1 and S2) when number concentrations were also higher. The higher CCN concentrations during IOP 2 drive more spatial variability in absolute differences between the 81 and 3 km cells as shown in Fig. S9. For CCN at 0.24% supersaturation, most of the 3 km cells are within 100 and 200 $cm^{-3}$ of the 81 km cell for IOPs 1 and 2, respectively. Most of the 3 km cells are within 200 and 300 $cm^{-3}$ of the 81 km cell for CCN at 0.46% supersaturation. Similarly, the higher CCN concentrations at 0.46% supersaturation produce broader distributions in the absolute differences than for CCN at 0.24% supersaturation.

**4.5 Spatial Representativeness of SGP Surface Aerosol Measurements**

The aircraft measurements during HI-SCALE provide an opportunity to assess the representativeness of the routine aerosol measurements collected at the surface from the Central Facility of the SGP site. Fig. 11 presents the temporal variability of ground OM, $SO_4$, $NO_3$, and $NH_4$ measurements collected by an AMS during HI-SCALE (Liu et al. 2021). These measurements are similar to those made by the operational Aerosol Chemical Speciation Monitor (ACSM) measurements (not shown). The gray shading denotes the range of the spatial variability one might expected over the 81 km cell around the SGP site. This range defined as the largest departures from zero for the 5th and 95th percentiles shown in Fig. 5, which is 41% for OM, 31% for $SO_4$, 61% for $NO_3$, and 62% for $NH_4$. The 81 and 3 km cell means within the boundary layer are denote by blue and red circles, respectively.

Interestingly, the 81 km cell means from the aircraft legs are often very similar to the Central Facility surface measurements for OM, $SO_4$, and $NH_4$. There are only a few instances in which the 81 km cell means fall outside the gray shading which would suggest that the surface measurements may not be representative over a large area. The right panels that plot the surface values averaged over the same time interval as the aircraft 81 km cell means show that most of the 81 km cell means close to the 1:1 line. The 3 km cells exhibit more spatial variability as expected, and most fall within the gray shading. However, there are days in which the 3 km cells have a broader distribution showing the estimate of spatial variability will not be reasonable on all days. In contrast, there are larger differences for $NO_3$, suggesting the variability of $NO_3$ around the SGP site is larger and that the estimates of spatial variability from the aircraft measurements are not as robust. In particular, the aircraft mean values are often higher than those at the SGP site.

Local precursor emissions are likely a significant source of spatiotemporal variability on $SO_4$ and $NO_3$. For example, emission rates of $SO_2$ from the nearby Redrock Power Plant and Ponca City Refinery (Fig. 1a) are relatively large that contribute to $SO_4$ formation in the region. The G-1 aircraft frequently intersected plumes of $SO_2$ and small particles nearby and downwind of these point sources. Despite the variability in $SO_2$ emissions and variable wind directions during IOP 1, the aircraft 81 km cell mean and the surface measurements of $SO_4$ are often quite similar (Fig. 11b). Emission precursors of $NO_3$, such as NOx and $NH_3$ are also highly variable over the SGP site which is reflected in the larger spatial variability in $NO_3$. The larger spatial variability may also be due to the shorter lifetime of $NO_3$ compared to $SO_4$ coupled with the local emission sources.

Figure 12 is similar to Fig. 11, except that the temporal variations from surface CPC3025, CPC 3010, SMPS, and CCN are shown. The gray shading denotes the range of spatial variability indicated by the maximum departures from zero for the 5th and 95th percentiles aircraft data shown in Fig. 8 which is 55%



for the CPC3025 and CP3010, 52% for SMPS, and 28% for CCN. Unfortunately, ground CCN data is not available prior to May 10 so that those measurements can only be compared to eight flights during IOP 1.

While the aircraft measurements reveal large spatial variations aerosol number concentrations around the SGP site as reflected by the range of 3 km cell means in Figs. 12a – 12c, it is interesting that that the 81 km cell means are frequently very similar to the surface measurements. The temporal variations in CCN shown in Fig. 12d are similar to the OM concentrations in Fig. 11a. As with the number concentrations, the aircraft 81 km cell means for CCN are similar to the ground measurements after May 10.

**5 Discussion**

When comparing observations with model predictions, it is important to note that "resolution" of spatial variability of an atmospheric variable and "grid spacing" have different meanings (Pielke et al. 1991; Grasso 2000). A minimum of four to six grid spacings are required to resolve variations in physical features, but the exact number is subjective (Durran 2000) and more grid spacings are preferable to fully

resolve those features. Nevertheless, many modeling studies still incorrectly use "resolution" and "grid spacing" interchangeably. In relation to the findings from this study, coarse climate models with grid spacings greater than a quarter of a degree will not adequately resolve the spatiotemporal variability in aircraft measurements of aerosol properties around the SGP site. However, their predictions of aerosol properties may agree reasonably well with aircraft measurements averaged over the coarse model grid

cells. Conversely, operational forecasting models and research chemical transport models with km-scale grid spacings will resolve much, but not all, of the spatiotemporal variability since a 3 km grid spacing can only resolve variations that are at a minimum four to six grid spacings wide (i.e., 12 to 18 km).

While the statistics shown in Section 4 describes spaital variability of aerosol properties in terms of all the flights over a month-long IOP, the amount of spatial variability in aerosol properties varied from day to

day. For example, some days had more or less spatial variability in aerosol composition as shown by the flight on May 7 in Fig. 3b. As shown in Fig. 11, the range of OM 3 km cell means around the SGP site was smaller on other days (e.g., April 25) and than on May 7. In contrast, May 7 also had little spatial variability in $SO_4$ compared to other days.

Aircraft sampling during HI-SCALE had many repeatable flight patterns over the same region to

characterize the day-to-day spatial variability, but the number of flights still can be considered limited to define typical spatial variability. Our results also indicate seasonal differences in spatial variability which may not be applicable to conditions during the fall and winter months. While there have been many aircraft missions of aerosol properties over the past several decades, many of them have not had repeatable transects over a small region that can be used to characterize spatial variability. For example,

flights during the Studies of Emissions and Atmospheric Composition, Clouds, and Climate Coupling by Regional Surveys (SEAC4RS, Toon et al. 2016) campaign were conducted over much of the continental U.S.; therefore, most of the measurements can only quantify spatial variability once over a specific region. Nevertheless, it may be possible to characterize spatial variability in the vicinity of Houston with the highest density of flight tracks where the research aircraft were based. Several campaigns have also

been conducted in Houston, including TexAQS 2000 and 2006 campaigns (Brock et al. 2003; Parish et al. 2009) among others, so that compiling measurements of aerosol properties from those campaigns may provide more robust estimates of variability that could also change in time as aerosol precursor emissions change. SENEX (Warneke et al. 2016) is another campaign that conducted grid-like aircraft flight paths over urban regions in the southeastern U.S.; however, most of the urban areas were sampled only once.

While subgrid-scale variability statistics similar to those in this study could be computed from those transects, the statistics would not be robust since they would depend on meteorological conditions and chemical evolution for a particular day.

It is also important to note that weather and climate models often have poor vertical resolution in addition to coarse grid spacings that impacts our concept of subgrid-scale variability. For example, large vertical



gradients in aerosol properties are frequently observed across the top of the boundary layer and lower troposphere which require small grid spacings on the order of tens of meters to resolve adequately. It is possible that aircraft measurements could be used to quantify subgrid-scale variability associated with vertical gradients; however, this aspect was not explored in this study since there were relatively few profiles made by the G-1 aircraft during HI-SCALE compared to the time spent on horizontal transects. In

addition, vertical profiles usually extended to just above the boundary layer top so that vertical gradients in aerosol properties across a large portion of the lower troposphere were not characterized. Other aircraft datasets may be more appropriate for this purpose (e.g. Wang et al. 2020). In the future, unmanned aerial platforms that can operate for longer periods of time may sufficient measurements to provide robust statistics on vertical gradients in aerosol properties.

**6 Summary and Conclusions**

In this study, we use aerosol properties measured by research aircraft during the 2016 HI-SCALE field campaign over north-central Oklahoma near the DOE ARM Southern Great Plains site to quantify their subgrid-scale variability over a range of spatial scales. Aerosol properties examined include bulk composition concentrations, composition derived from a single particle mass spectrometer, aerosol

number concentrations, aerosol size distribution, and CCN concentrations. To compute subgrid-scale variability statistics, the aircraft flight paths are divided into segments that fall within grid cells that are 81, 27, 9, and 3 km wide. The coarser and finer grid spacings are consistent with those used by current global climate and operational forecasting models, respectively. Only the constant altitude flight legs within the convective boundary layer are used to minimize variations associated with the vertical

gradients in aerosol properties. Day-to-day variations in mean aerosol properties are also removed from the statistics to isolate variations due solely to spatial variability. Multiple aircraft flights occurred over the same region, so that more robust statistics of subgrid-scale variability could be obtained. The seasonality of subgrid-scale variability in aerosol properties is quantified by comparing the statistics from the spring and late-summer periods.  In addition, we explore the representativeness of surface point

measurements at the SGP site by comparing the variability of aerosol properties in the boundary layer with those at the ground.

Not surprisingly, we find that the coarse 81 km cell averages miss substantial variability in aerosol composition, mixing state, number, size distribution, and CCN over the SGP site. Even though this site is in a rural area, relatively large variations in aerosol properties are produced because of local variations in

anthropogenic and biogenic precursor emissions and chemical formation superimposed on background aerosols transported from upwind regions. We also show that:

- The subgrid-scale variability associated among the 27 km cells is qualitatively similar to the distribution obtained from the 9, and 3 km cells. This does not imply that global climate models with ~0.25 degree grid spacings are capable of resolving the observed aerosol variability at the SGP

because a minimum of four to six grid cells are needed to resolve spatial variations in atmospheric constituents. Therefore, smaller grid spacings consistent with current operational forecasting models are likely needed to explicitly represent the observed spatial variability in aerosol properties.

- As shown in Fig. 5, 90% of the 27, 9, and 3 km cell mean OM concentrations varied by as much as 46% of the 81 km cell mean around the SGP site. Similar statistics for $SO_4$ and $NO_3$ produced lower

(as much as 35% different) and higher (as much as 64% different) spatial variability, respectively, than OM.

- 90% of the 27, 9, and 3 km cell mean accumulation mode number concentrations varied from the 81 km mean by as much as 35%. The spatial variability in number concentrations that included ultrafine and Aiken mode particles were larger, producing variations as large as 68% of the 81 km cell mean.



Much of this variability was due to local sources of ultrafine particles close to and downwind of the
        Redrock Power Plant and Ponca City Refinery and more widespread NPF events during IOP 1.

- The spatial variability of some composition species and particles less than 50 nm in diameter have skewed distributions.

- Analysis of the variations in size distributions (Fig. 9) reveal that the 81 km cell mean is often quite
different from variability of the 3 km cell means over the SGP site. This suggests that coarse climate models may misrepresent CCN concentrations because subgrid scale variability in size distribution is neglected. Averaging observed size distribution variations can also hide bimodal distributions that reflect chemical processes contributing to particle growth.

- Even the 3 km cell averages miss extreme local maxima and minima in aerosol properties, especially
for number concentrations of small particles.

- Even though there are large variations in aerosol properties around the rural SGP site, we find the 81 km averages of the aircraft properties are often very close to the ground measurements made at a single point. The exception is $NO_3$, but since $NO_3$ concentrations are usually low during HI-SCALE those differences do not seem to significantly impact comparisons of average aircraft and surface
aerosol number and CCN.  This suggests that coarse global model predictions of aerosols properties can be directly compared to SGP site measurements. However, it would be useful to account for the range of observed variability derived from the aircraft measurements to indicate how much local variability the model does not represent, similar to accounting for measurement uncertainties that complicate measurement-to-model comparisons.

In terms of seasonal differences in subgrid-scale variability, we found that:

- Spatial variability in OM in terms of absolute differences with the 81 km mean (Fig. S5) increased from spring to late summer than during the spring, but the percentage differences (Fig. 5) had smaller seasonal dependance. This suggests that the higher concentrations themselves during IOP 2 contributed to the increased variability of absolute differences. The higher OM concentrations during
the late summer likely resulted from higher SOA formation rates coupled with variability in both anthropogenic and biogenic precursor sources.

- In contrast with OM, both absolute concentrations (Fig. S5) and percentage differences (Fig. 5) of $NO_3$ had more variability during IOP 1. The lower variability during IOP 2 was due to both lower concentrations and more uniformly distributed $NO_3$.

- Analysis of the composition classes derived from a single particle mass spectrometer suggest more complex seasonal differences than bulk composition measurements. While there were relatively small differences between IOP 1 and 2 for many mixtures of organics, fresh OM (Org1) and aged OM (Org 2) had less spatial variability during the summer. During IOP 1, the majority of organics were from biomass burning or were mixed with sulfate and nitrate with relatively lower and spatially more
variable concentrations of pure organic-rich particles. During IOP 2, OM was composed primarily of organic-rich particles that were less variable over the SGP site.

- Consistent with the bulk OM composition measurements, the spatial variability the absolute differences in mean accumulation mode aerosol number (Fig. S8) between the 27, 9, and 3 km cells and the 81 km cell mean was larger during the late summer while the percentage differences (Fig. 8)
did not vary much between IOP 1 and 2. Even though IOP 1 had more frequent NPF events, the spatial variability in ultrafine and Aiken mode particles was larger during IOP 2. While there were large variations in total number concentrations during IOP1, average concentrations were also higher in general during IOP 1 so that the overall spatial variability was larger during IOP 2.

- Seasonal differences in spatial variability of CCN was also consistent with the variability associated
with bulk composition and accumulation mode aerosol number, with little seasonal differences in



terms of percentage differences with the 81 km cell mean (Fig. 10) and larger spatial variability in the absolute differences during IOP 2 (Fig. S10).

The methodology of computing subgrid-scale aerosol variability statistics could be extended to other aircraft campaigns with repeated flight tracks over a particular region. For example, past G-1 aircraft
missions over Mexico City (Kleinman et al. 2008), Cape Cod (Berg et al. 2016), Sacramento (Zaveri et al. 2012), and downwind of Manaus Brazil (Shilling et al. 2018) are potential candidates that can be used to quantify subgrid-scale variability directly over and just downwind of large urban areas. While the magnitude of aerosol concentrations and the footprint from urban sources likely change over the years due to growth and/or emission controls, these datasets would provide an estimate of spatial variability
influenced by meteorological conditions that are likely similar from year to year. For aerosols in a more remote continental region, the recent Cloud, Aerosol, and Complex Terrain Interactions (CACTI) field campaign (Varble et al. 2021) provided repeated aerosol measurements over a limited area that would be useful to assess how subgrid-scale variabilities in aerosol properties influence the lifecycle of deep convection.

In addition to providing alternative methods of evaluating model predictions, such information derived from aircraft field campaigns could also be used to develop numerical treatments that consider subgrid-scale aerosol variability on aerosol-cloud-radiation interactions in conjunction with existing treatments of cloud parameterizations.

**Data Availability**

Data used in this manuscript are available from the ARM data archive (www.arm.gov/data and www.arm.gov/research/campaigns/sgp2016hiscale).

**Supplement**

**Author Contributions**

JDF performed the data analyses and prepared he manuscript with contributions from JS, JL, AZ, DB,
KS, FM, JT, GS, and RZ. JS and JL are the mentors of the AMS instrument, AZ, DB, and KS are mentors of the miniSPLAT instrument, JW is the instrument mentor of the FIMS instrument, and FM and JT are mentors of the CPC and CCN instrumentation on the G-1 aircraft. GS contributed to the analysis producing the particle class information based on the miniSPLAT measurements.

**Acknowledgements**

The HI-SCALE field campaign was supported by the Atmospheric Radiation Measurement (ARM) Climate Research Facility and the Environmental Molecular Sciences Laboratory (EMSL) through projects 48804 and 49297, both are U.S. Department of Energy (DOE) Office of Science User Facilities sponsored by the Office of Biological and Environmental Research. The ground deployment of an HR-ToF-AMS was supported by EMSL. We thank Tamara Pinterich for operating the FIMS instrument
during HI-SCALE. This research was supported by the Atmospheric System Research (ASR) program as part of the DOE Office of Biological and Environmental Research under PNNL project 57131. Pacific Northwest National Laboratory is operated by DOE by the Battelle Memorial Institute under Contract DE-A06-76RLO 1830.




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


Table 1. Aerosol instrumentation deployed on the G-1 aircraft during HI-SCALE

| Measurement | Sampling interval | Instrument |
|---|---|---|
| Total aerosol number | 1 second | TSI: Condensation Particle Counter (CPC) 3010 (>10 nm) and 3025 (>3 nm), one CPC 3010 behind isokinetic inlet and another behind the CVI inlet |
| Aerosol size distribution | 1 second | Brookhaven National Laboratory (BNL): Fast Integrated Mobility Spectrometer (FIMS) (9 – 426 nm), either isokinetic or CVI inlet |
| Cloud condensation nuclei (CCN) | 1 second | DMT: CCN counter at two super-saturations |
| Aerosol composition (bulk) | 13 seconds | Aerodyne: High-resolution time-of-flight aerosol mass spectrometer (HR-ToF-AMS), either isokinetic or CVI inlet |
| Aerosol Composition (single particle) | 60 seconds | PNNL: miniSPLAT, either isokinetic or CVI inlet |

Table 2. Particle classes defined from miniSPLAT single particle measurements.

| Name | Description | used in this study |
|---|---|---|
| Soot | soot containing particles | yes |
| Sulfate_org | sulfate, nitrate, and organic mixed particles | yes |
| Nitrate_org | nitrate and organic mixed particles | yes |
| Org1 | fresh, organic rich particles | yes |
| Org2 | aged (oxygenated), organic rich particles | yes |
| BB | fresh biomass burning particles | yes |
| BB_SOA | biomass burning particles showing markers of aged organics | yes |
| Org_amines | organic particles with amine function group markers | no |
| IEPOX_SOA | secondary organic aerosol containing isoprene epoxydiol markers | yes |
| Dust | dust particles | yes |
| Pyr | particles containing the pyridinium ion (mostly in free troposphere) | no |



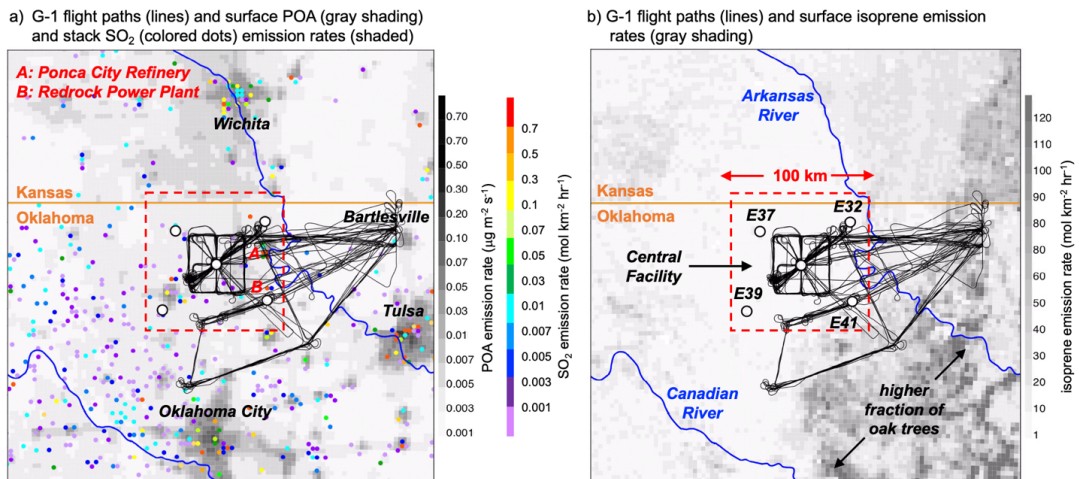

Figure 1.    G-1 flight paths near the ARM SGP site over north-central Oklahoma during IOP 2 along with a) surface POA (gray shading) and stack SO$_2$ (colored dots) emissions and b) surface isoprene emission rates (gray shading). Open circles denote primary SGP site vertical profiling measurement sites.

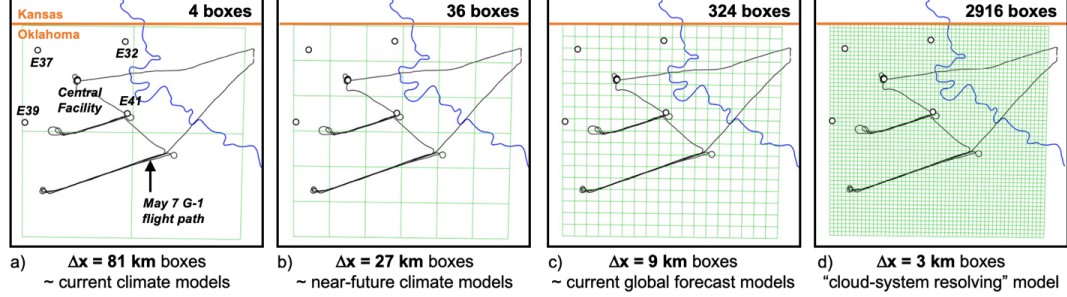

Figure 2.    G-1 aircraft flight path on May 7 along with the a) 81, b) 27, c) 9, and d) 3 km boxes used to compute mean aerosol properties from the aircraft measurements.

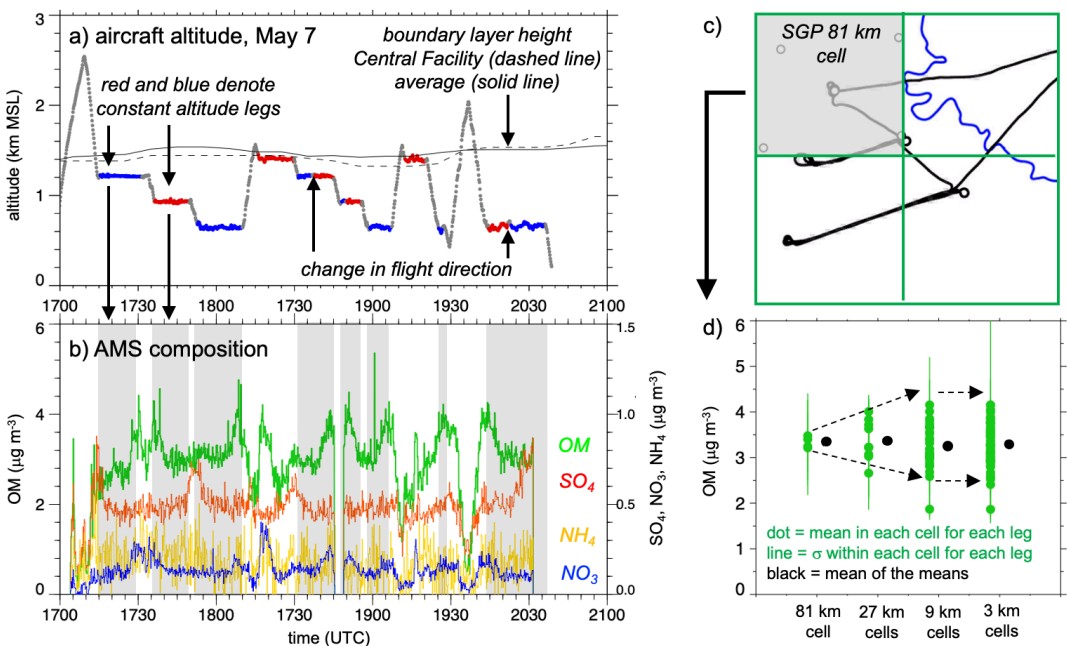

Figure 3.    G-1 aircraft a) altitude on May 7 where the alternating blue and red colors denote different constant altitude flight legs and b) concentrations of OM, SO$_4$, NO$_3$ and NH$_4$ measured by the AMS where gray shading denotes boundary layer flight legs used to compute subgrid-scale variability. The gray shading in c) denotes the 81 km cell around the SGP site associated with the 81, 27, 9, and 3 km cell averages and standard deviation of OM shown in d).

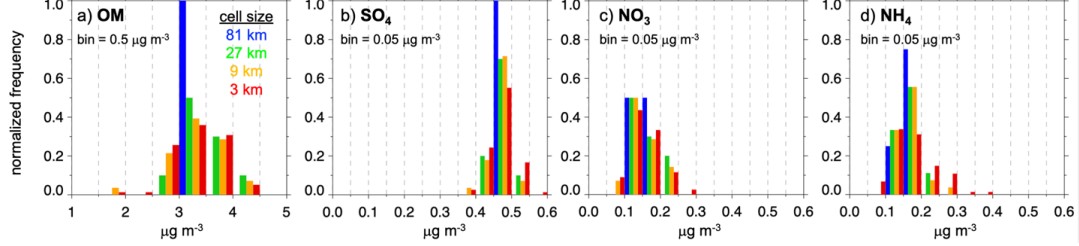

Figure 4.    Normalized frequency of occurrence for a) OM, b) SO$_4$, c) NO$_3$, and d) NH$_4$ concentrations observed for the May 7 flight shown in Figure 3b.





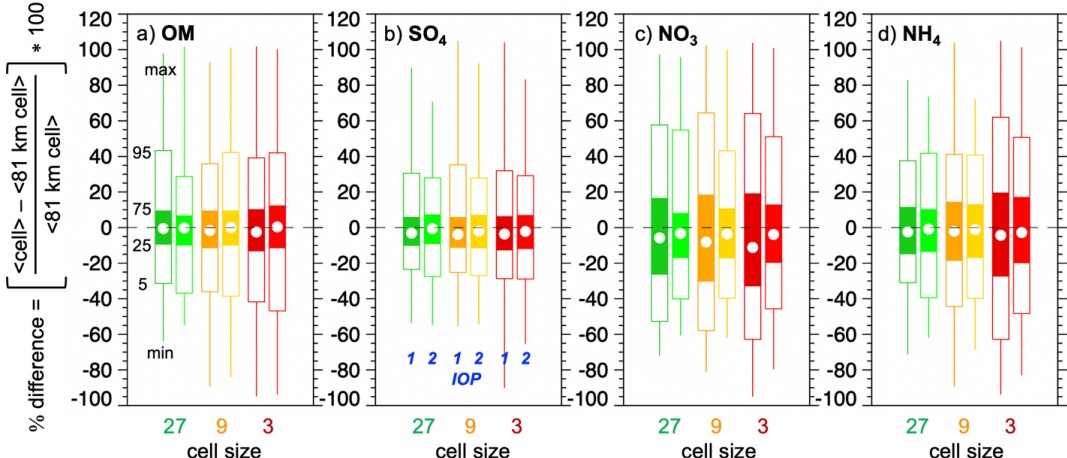

Figure 5. Percent difference between the SGP 81 km cell average and the 27, 9, and 3 km cell averages of a) OM, b) SO$_4$, c) NO$_3$, and d) NH$_4$ concentrations from the AMS instrument expressed as percentiles where the white circles denote the median values. All flights during IOP 1 and 2 are grouped by the darker (left) and lighter (right) colors, respectively.

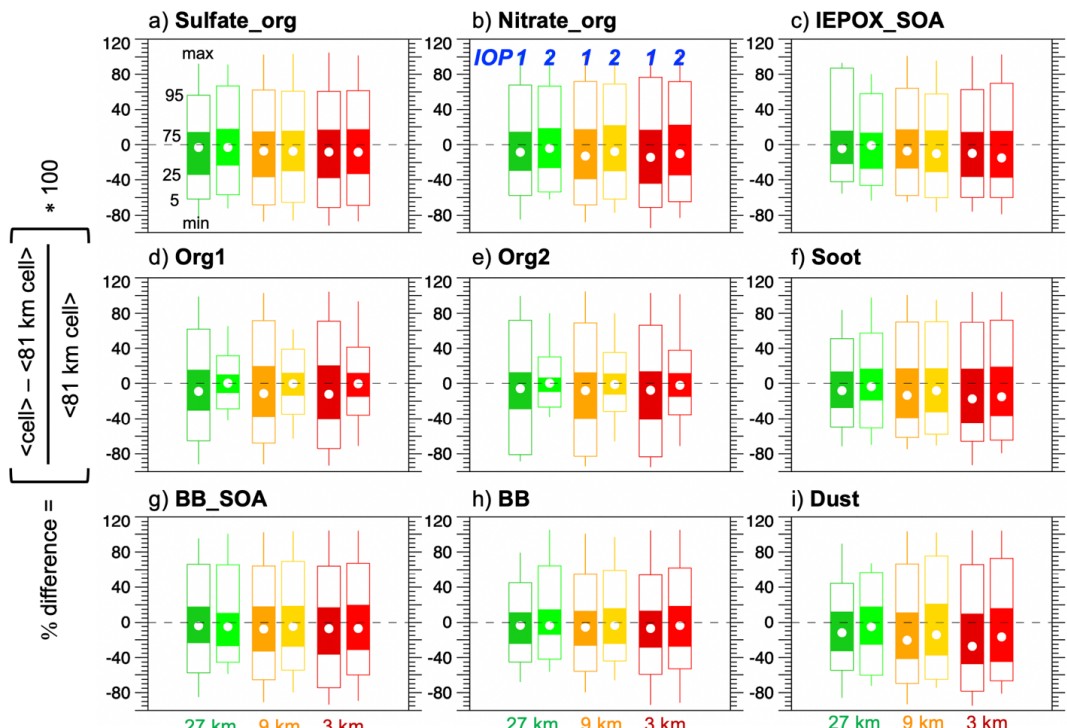

Figure 6. Percent difference between the SGP 81 km cell average and the 27, 9, and 3 km cell averages of a) Sulfate_org, b) Nitrate_Org, c) IEPOX_SOA, d) Org1, e) Org2, f) soot, g) BB_SOA, h) BB, and i) dust volume concentration classes from the miniSPLAT instrument expressed as percentiles where the white circles denote the median values. All flights during IOP 1 and 2 are grouped by the darker (left) and lighter (right) colors, respectively.


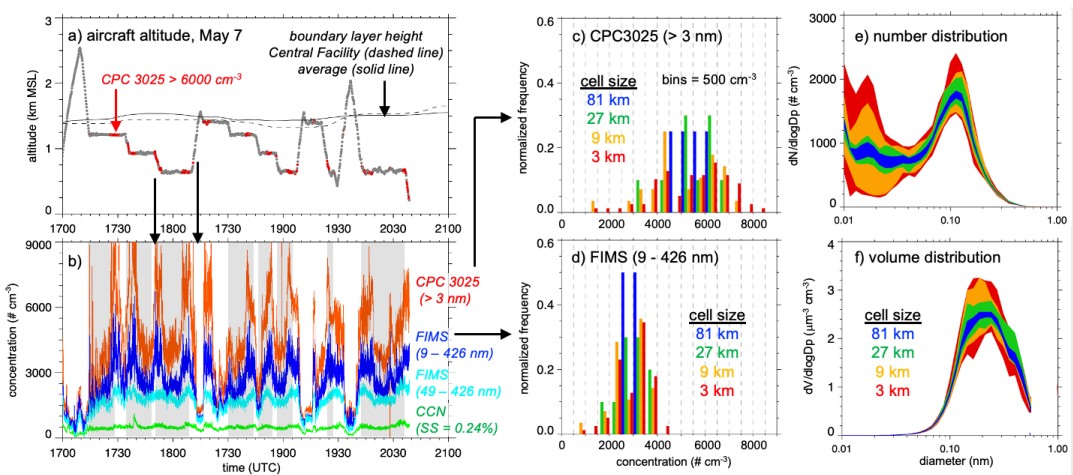

Figure 7.    G-1 aircraft a) altitude on May 7 where the alternating red colors denote locations of peak number concentrations and b) number concentrations from the CPC3025, FIMS, and CCN instruments where the gray shading denotes boundary layer flight legs used to compute subgrid-scale variability. Normalized frequency from the c) CPC3025 and b) FIMS instruments and the e) number and f) volume distribution associated with the 81, 27, 9, and 3 km cell averages within the 81 km cell around the SGP site.

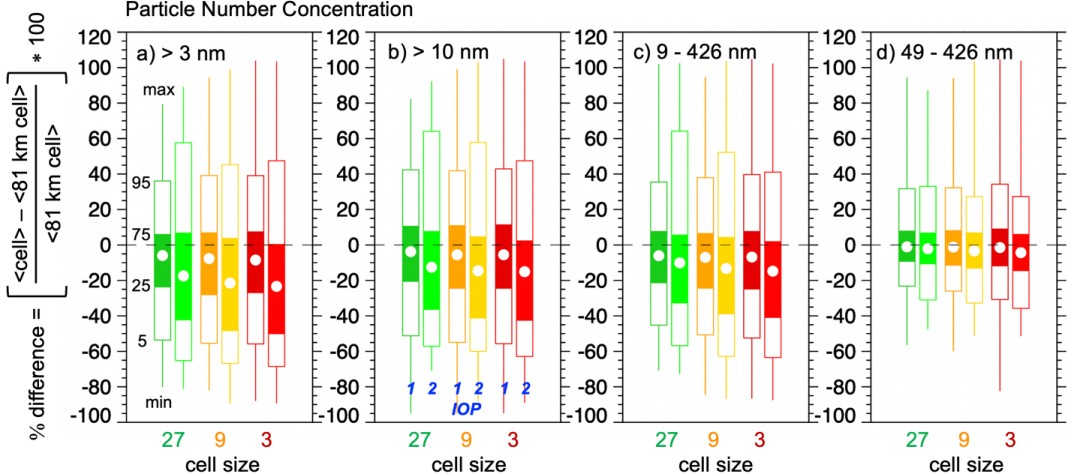

Figure 8.    Percent difference between the SGP 81 km cell average and the 27, 9, and 3 km cell averages of for number concentrations with particle diameters a) > 3 nm (CPC3025), b) > 10 nm (CPC3010), c) 9 – 426 nm (FIMS), and d) 49 - 426 nm (FIMS) expressed as percentiles where the white circles denote the median values. All flights during IOP 1 and 2 are grouped by the darker (left) and lighter (right) colors, respectively.





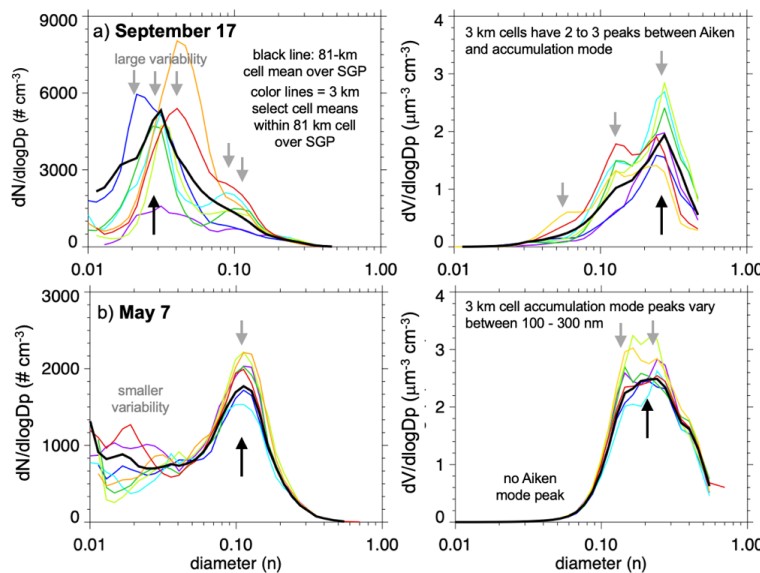

Figure 9.    Aerosol number and volume size distributions on a) September 17 and b) May 7. Black line
denotes the average within the 81 km cell over the SGP site and the color lines denote select
3 km cell averages within the same 81 km cell.

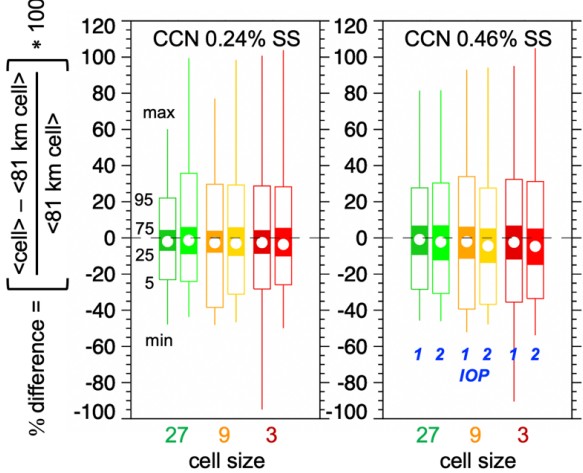

Figure 10.   Percent difference between the SGP 81 km cell average and the 27, 9, and 3 km cell averages
of CCN at a) 0.24% supersaturation and b) 0.46% supersaturation expressed as percentiles
where the white circles denote the median values. All flights during IOP 1 and 2 are grouped
by the darker (left) and lighter (right) colors, respectively.



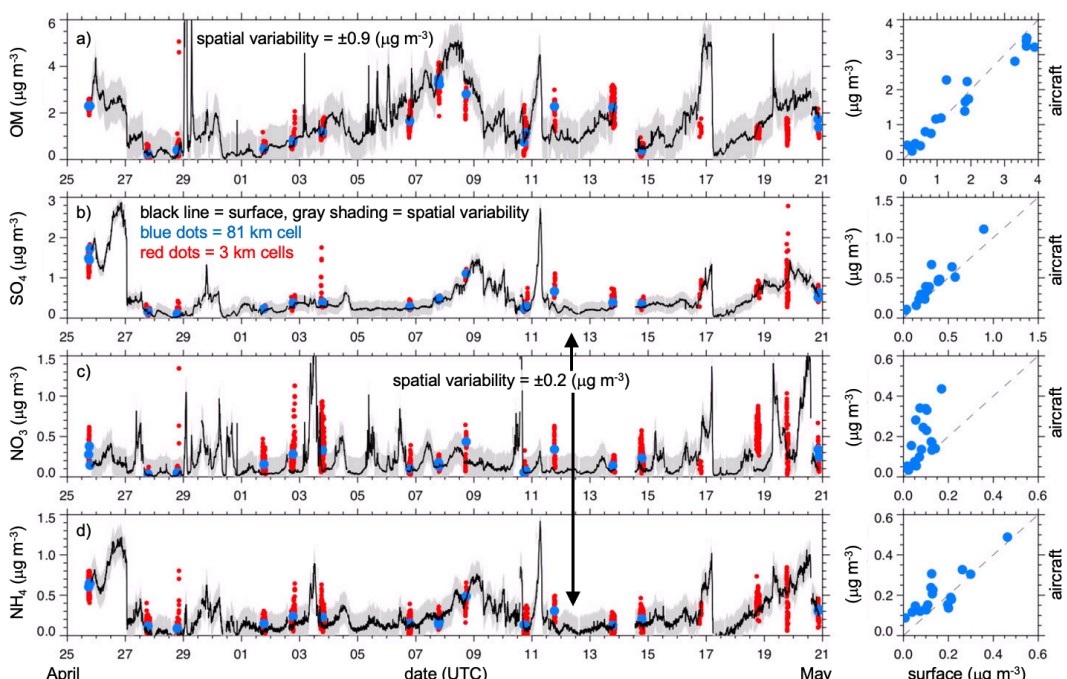

Figure 11. Measured a) OM, b) SO$_4$, c) NO$_3$ and d) NH$_4$ at the surface SGP site during IOP 1 (black lines), along with aircraft 81 km cell averages within the boundary layer (blue dots) and 3 km cell averages (red dots) with the 81 km cell over the SGP site. Right panels compare coincident in time surface and aircraft measurements.

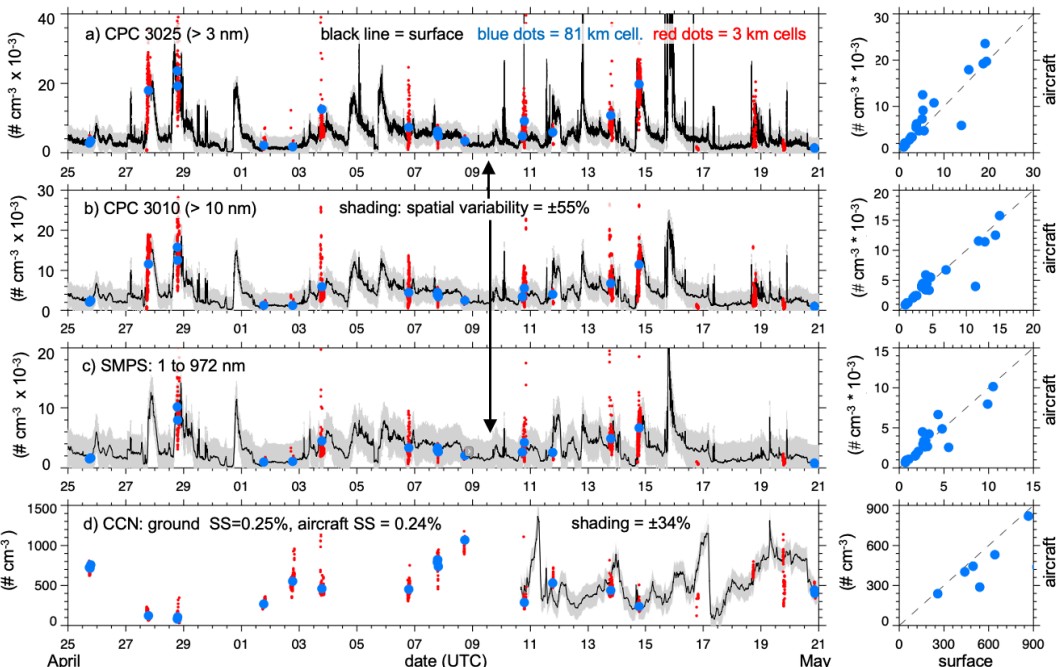

Figure 12. Measured a) number concentration > 3 nm, b) number concentration > 10 nm, c) number concentration between 1 and 972 nm, and d) CCN concentration at the surface SGP site during IOP 1 (black lines), along with aircraft 81 km cell averages within the boundary layer (blue dots) and 3 km cell averages (red dots) with the 81 km cell over the SGP site. Right panels compare coincident in time surface and aircraft measurements.
