# Peer review of "Using Aircraft Measurements to Characterize Subgrid-Scale Variability of Aerosol Properties Near the ARM Southern Great Plains Site"

_Atmospheric Chemistry and Physics, 2022_

## Referee Comment (RC2)

Review of 'Using Aircraft Measurements to Characterize Subgrid-Scale Variability of Aerosol Properties Near the ARM Southern Great Plains Site'

This study uses the aircraft in-situ measurements from the HI-SCALE campaign to characterize the sub-grid variability of aerosols over the ARM-SGP site, by applying the methodology of comparing the averaged aerosol properties in the scales of 81, 27, 9, and 3 km. Results show substantial variabilities in aerosol composition, concentration, size distribution, as well as CCN over the SGP. Hence, the model misrepresentation of the sub-grid variabilities might induce uncertainties in simulating the aerosol direct and indirect forcings. The authors present robust and statistically sound analysis within one field campaign, and also express the expectation of further application of the methodology. I recommend publication after a few minor comments of mine are considered and addressed.

**Minor Comments:**

**Line 157.** The TSI 3010 was designed for measured particle concentration up to 10000/cm-3 with very little coincidence. However, in Table S1 there are 4 cases that have averaged CPC > 10000/cm-3. How would you reconcile those results, whether they are comprised of real signals or noises (e.g., instrument glitch, cloud water splash...)? Please give a discussion on it.

TSI, Model 3010 Condensation Particle Counter Instruction Manual, https://ethz.ch/content/dam/ethz/special-interest/usys/iac/iac dam/documents/edu/courses/atmospheric_physics_lab_work/TSI-3010.pdf, 2022.

**Line 236.** 'excluded from…'?

**Line 261.** Readers might also be interested in seeing the relationships between those departure values versus the relative position of the targeted cell within that 81 km grid box (e.g., departures at southeast grid box versus northwest grid box), under different background wind conditions. Perhaps some empirical functions can be deduced from those relationships, if any.

**Line 282.** Is there any potential explanation for the relatively larger spatial variabilities in $NO_3$ and $NH_4$?

**Line 291.** What about NH4?

**Line 293.** Is the bi-modal distribution of IOP2 $SO_4$ a reflection of any cloud processing signals (e.g., in-cloud sulfate production), or purely due to the local emission variabilities?

**Line 389.** '81, 27, 9, and 3 km...'

**Line 390.** Is there a redundant preposition here?

**Line 402.** Could you explain why the CPC 3025/3010/FIMS ranges (5-95) for 27 km are generally higher than the 9km and 3km in IOP2? Seems counter-intuitive compared to previous figures.

**Line 404.** Please define NPF before using it.

**Line 696.** 'prepared the manuscript...'

**Section 4.4.** Since you have the simultaneous measurements of aerosol and CCN at two supersaturation levels, it would be interesting to see if the spatial variabilities of the aerosol hygroscopicity (or activation capacity) share similar relationships to either aerosol or CCN.

**Figure 5.** Can you also define the meaning of the box edges and whisker in the caption? And please add the mean values, maybe as short lines?

---

## Author Comment (AC1)

Response to Reviewer 1:

Reviewer comments in black, response in blue

This study reports an analysis of airborne measurements over the Atmospheric Radiation Measurement (ARM) program's Southern Great Plains (SGP) site during the Holistic Interactions of Shallow Clouds, Aerosols and Land Ecosystem (HI-SCALE) campaign, with a focus on sub-grid scale variability. This is of course important as it relates to model applications as models typically assume homogeniety in aerosol properties in a grid cell. This study wisely quantifies subgrid variability iin terms of both normalized frequency distributions and percentage difference percentiles using grid spacings of 3, 9, 27, and 81 km; the rationale for this spacing choices is that they represent those typically used by cloud-system resolving models as well as the current and next generation climate models.

As someone involved with many field campaigns, I found this study to be very refreshing and useful. The results are significant showing large horizontal gradients for this rural location. Number concentrations were shown to be quite variable owing to events such as nucleation. The degree of spatial variation was shown to vary seasonally. Aircraft measurements were in similar for many (but not all) aerosol properties measured at the ground SGP site. An application of the findings is that the reported variability from the airborne data can serve as an uncertainty range when comparing the surface data to model predictions that rely on coarse grid spacings. I recommend publication subject to minor corrections below which are all of mostly an editorial nature. The science and analysis was robust and i do not have much to add in that category.

Response: We thank the reviewer for these positive comments, which includes pointing out the following typos and suggested clarifications that have now been resolved.

~Lines 134-140: state year of measurements by day/month info.

Response: Changed date ordering to day/month.

Line 212: "cell" should probably be plural

Response: Changed "cell" to "cells".

Line 215: what criteria were used for knowing when data were not contaminated by cloud?

Response: The G-1 flight team provided a cloud flag as part of the data files that is based on cloud liquid water content reaching a threshold of 0.001 g/m3. Other cloud probes were used for cross-checking this flag. The text now includes a phrase about the cloud flag.

Line 223: should be "…increases for the…"

Response: Fixed tense.

Line 233-235: it would be informative to know exactly what criteria were applied to do this separation within and above the BL.

Response: The BL heights from the Doppler lidar are determined by the Tucker method (Tucker et al. 2009; Krishnamurthy et al. 2021). For situations in which the Doppler lidar data were not available and

we used soundings, BL heights were estimated visually using the aircraft measurements and radiosonde profiles that occurred close to the flight time.  A specific threshold for vertical potential temperature or humidity gradient was not used.  The text has been modified to include the citations and additional description.

---

## Author Comment (AC2)

Response to Reviewer 2:

Reviewer comments in black, response in blue

This study uses the aircraft in-situ measurements from the HI-SCALE campaign to characterize the sub-grid variability of aerosols over the ARM-SGP site, by applying the methodology of comparing the averaged aerosol properties in the scales of 81, 27, 9, and 3 km. Results show substantial variabilities in aerosol composition, concentration, size distribution, as well as CCN over the SGP. Hence, the model misrepresentation of the sub-grid variabilities might induce uncertainties in simulating the aerosol direct and indirect forcings. The authors present robust and statistically sound analysis within one field campaign, and also express the expectation of further application of the methodology. I recommend publication after a few minor comments of mine are considered and addressed.

Response: We thank the reviewer for the suggestions and pointing out the following typos have now been resolved.

Minor Comments:

Line 157. The TSI 3010 was designed for measured particle concentration up to 10000/cm-3 with very little coincidence. However, in Table S1 there are 4 cases that have averaged CPC > 10000/cm-3. How would you reconcile those results, whether they are comprised of real signals or noises (e.g., instrument glitch, cloud water splash...)? Please give a discussion on it.

TSI, Model 3010 Condensation Particle Counter Instruction Manual,

https://ethz.ch/content/dam/ethz/special-interest/usys/iac/iac
dam/documents/edu/courses/atmospheric_physics_lab_work/TSI-3010.pdf, 2022.

Response: The G-1 team applied the coincidence correction to the CPC data (Aaron et al. 2013). The coincidence correction extends the CPC limit to $8*10^4$ cc with an average discrepancy of less than 4%. The discrepancy is less than 15%, when the concentration is under $5*10^5$ #/cc. Thus, most of the HI-SCALE data is valid. A sentence has been added to mention the coincidence correction.

Line 236. 'excluded from...'?

Response: Changed as suggested.

Line 261. Readers might also be interested in seeing the relationships between those departure values versus the relative position of the targeted cell within that 81 km grid box (e.g., departures at southeast grid box versus northwest grid box), under different background wind conditions. Perhaps some empirical functions can be deduced from those relationships, if any.

Response: This is an interesting idea. We plotted up the departure values as a function of cells within the 81 km cell (or all four 84 km cells) for a few cases; however, it quickly became apparent that such an analysis will generate many figures for the different variables, even if they are consolidated into each IOP. On some days it will be evident that local emission sources may drive the local spatial variability, but on other days longer-range transport that is outside of the sampling area will drive the spatial variability (or lack thereof). Therefore, we believe that such an analysis is best left to other studies to keep the present paper focused on quantification of the spatial variability.

Line 282. Is there any potential explanation for the relatively larger spatial variabilities in NO3 and NH4?

Response: For NO3, part of the answer is given in the next paragraph, i.e. the second paragraph of Section 4.1. There are fewer large spikes in NO3 time series compared to OM, and those NO3 spikes are relatively larger than the background values compared to OM. For example, in Figure 3b there is one NO3 spike which is four times higher than the background value, but there are no OM spikes that depart from the background by that much. For NH4, the spatial variability is likely affected by a noisier signal from the AMS than from OM, SO4, and NO3, which warrants some caution regarding the variability of NH4.

Line 291. What about NH4?

Response: We originally left off NH4 since the change in mean concentrations between the IOPs was small; however, this has been added for completeness.

Line 293. Is the bi-modal distribution of IOP2 SO4 a reflection of any cloud processing signals (e.g., in-cloud sulfate production), or purely due to the local emission variabilities?

Response: To answer this question would most likely the require the use of modeling to differentiate the effect of individual processes such as clouds and emissions on the bi-modal distribution, which is beyond the objectives of this study. There were far fewer days with clouds during IOP 2, so the bi-modal distribution may be due to other factors.

Line 389. '81, 27, 9, and 3 km...'

Response: Fixed typo.

Line 390. Is there a redundant preposition here?

Response: Deleted "with"

Line 402. Could you explain why the CPC 3025/3010/FIMS ranges (5-95) for 27 km are generally higher than the 9km and 3km in IOP2? Seems counter-intuitive compared to previous figures.

Response: This is a good point. We do not have an explanation for this result, but we have added a sentence to point this out to readers. Since this trend does not occur with the accumulation mode size particles, it likely has something to do with ultrafine particles and/or new particle formation and the spatial extent of ultrafine plumes that may differ between IOP 1 and 2.

Line 404. Please define NPF before using it.

Response: Defined NPF.

Line 696. 'prepared the manuscript...'

Response: Changed "he" to "the".

Section 4.4. Since you have the simultaneous measurements of aerosol and CCN at two supersaturation levels, it would be interesting to see if the spatial variabilities of the aerosol hygroscopicity (or activation capacity) share similar relationships to either aerosol or CCN.

Response: We have changed Figures 10 and S9 to now include panels for aerosol hygroscopicity. Two paragraphs have been added to Section 4.4 that describe how hygroscopicity is calculated and discuss spatial variability of aerosol hygroscopicity. Since Gouihar Kulkarni computed the kappa values for another paper on CCN closure, he has been added to the co-author list.

Figure 5. Can you also define the meaning of the box edges and whisker in the caption? And please add the mean values, maybe as short lines?

Response: Figure caption has been changed to define the meaning of the percentiles. Added the mean values as horizontal black lines. For consistency, we added the mean values to all similar figures that use percentiles.